# The Impact of Urbanization on Taxonomic Diversity and Functional Similarity among Butterfly Communities in Waterfront Green Spaces

**DOI:** 10.3390/insects14110851

**Published:** 2023-10-31

**Authors:** Wenqiang Fang, Xiaoqian Lin, Ying Lin, Shanjun Huang, Jingkai Huang, Shiyuan Fan, Chengyu Ran, Emily Dang, Yuxin Lin, Weicong Fu

**Affiliations:** 1College of Landscape Architecture and Art, Fujian Agriculture and Forestry University, 15 Shangxiadian Rd., Fuzhou 350000, China; 1211775012@fafu.edu.cn (W.F.); 1221775002@fafu.edu.cn (X.L.); linyy_1118@163.com (Y.L.); hsj7174@163.com (S.H.); 13375002763@163.com (J.H.); 5221726035@fafu.edu.cn (S.F.); yeran190307@163.com (C.R.); 1221775031@fafu.edu.cn (Y.L.); 2Engineering Research Center for Forest Park of National Forestry and Grassland Administration, 15 Shangxiadian Rd., Fuzhou 350002, China; 3Faculty of Forestry, The University of British Columbia, Vancouver, BC V6T 1Z4, Canada; emily.dang@hotmail.com

**Keywords:** activity space group, butterfly diversity, diet group, function diversity, indicator species, urban waterfront green space, α and β diversity

## Abstract

**Simple Summary:**

In the era of expanding and enhancing urbanization, the conservation of urban biodiversity has gradually become a research hotspot. As an excellent indicator species reflecting the quality of the ecological environment, butterflies can be used to monitor and improve the quality of habitats. We explored the effects of urbanization on the α-diversity, β-diversity, and functional diversity of butterflies and analyzed the indicative species of different ecological gradients. This examination was aimed at understanding the effects of urbanization on the taxonomic diversity and functional similarity of butterflies and proposing scientific suggestions and strategies to improve the ecological quality of urban environments.

**Abstract:**

Urbanization has been shown to cause biodiversity loss. However, its effects on butterfly taxonomic and functional diversity still need to be studied, especially in urban waterfront green spaces where mechanisms of impact still need to be explored. We used butterflies as indicators to study how urbanization affects their taxonomic and functional diversity and identify indicator species in different urban ecological gradient areas. From July to September 2022, we surveyed 10 urban waterfront green spaces in Fuzhou City, China. We recorded 1163 butterflies of 28 species from 6 families. First, we explored the effects of urbanization on butterfly communities and made pairwise comparisons of different urban ecological gradients (α-diversity); secondly, we looked for differences between butterfly communities across urban ecological gradients (β-diversity); finally, we investigated differences in the response of butterfly functional groups to different urban ecological gradient areas and identified ecological indicative species. This study found the following: (1) Urbanization has led to the simplification of butterfly community structure, but there are also favorable factors that support the survival of individual butterflies; (2) Urbanization has led to significant differences in butterfly communities and plant-feeding polyphagous butterfly groups; (3) Urbanization has led to differences in the functional diversity of butterfly diet and activity space groups; (4) We identified five eco-indicator species in different urban ecological gradients.

## 1. Introduction

### 1.1. The Impact of Rapid Urbanization

Global environmental change is exerting significant pressure on species survival, with rapid urbanization emerging as one of the primary contributors to the decline in urban biodiversity [1]. This is in addition to habitat fragmentation and the degradation of natural ecosystems caused by urbanization, which also alters urban population densities, build-up densities, climate cycles, land-use patterns, and plant composition, directly or indirectly triggering a decline in urban biodiversity [2,3,4,5]. However, urbanization is a process that radiates from urban centers to the peripheries. Its impact beyond urban areas also affects rural regions, thereby creating ecological zones of varying ecological quality [6,7]. This phenomenon results in notable disparities in the distribution of biological species and populations [8]. Certain species or populations have the capacity to adapt and flourish in highly urbanized environments, while others are vulnerable, managing to survive outside city centers or within high-quality patches within urban environments. This phenomenon implies the presence of ‘environmental stress’ [9].

### 1.2. The Ecological Role of Butterflies

Butterflies, one of the most common pollinating insects in cities [10], are highly sensitive to environmental changes due to their susceptibility to surrounding temperatures and short life cycles [11]. They establish close interdependence and co-evolution with plants for reproduction and survival. Beirao and Soga et al. (2021 and 2015) have demonstrated a positive correlation between butterfly and plant diversity. The more diverse the plant life, the greater the diversity of butterflies within a specified range [12,13]. When urbanization impacts the environment, butterflies exhibit rapid responses, rendering them valuable indicators for monitoring ecological conditions and habitat quality. They have been proven to be excellent models for studying population dynamics and ecology [14]. However, there may be significant differences in how butterfly species respond to environmental changes, resulting in variations in butterfly taxonomy and function.

### 1.3. Status and Purpose of the Study

Although urbanization has been emphasized as one of the major driving factors of biodiversity endangerment, there is still a lack of research on the impact of urbanization on biodiversity in many countries, especially in developing nations [15]. China, the world’s largest developing country, is experiencing rapid urbanization. Therefore, studying the impact of urbanization on butterflies in China is of paramount importance. On an international scale, research conducted by urban ecologists primarily focuses on taxonomy, with fewer studies examining the impact on butterfly functional ecology [16,17]. 

Therefore, this study aims to explore the taxonomic and functional diversity of butterflies in response to various levels of urbanization in order to mitigate the negative impacts of urbanization on these insects. Building on prior research, we used α and β diversity to explore butterfly communities’ and functional groups’ responses to waterfront green space in different urban ecological gradient areas [18]. Furthermore, we evaluated the functional diversity of butterflies in various ecological gradient areas and identified indicative species to monitor ecological changes across different levels of urbanization. Understanding how urbanization affects butterfly diversity will provide insights for devising strategies to conserve urban biodiversity and improve the environmental quality of urban areas. To achieve this, we propose three hypotheses: (1) Urbanization may impact butterfly communities due to variations in ecological quality across different urban gradients; (2) The richness and abundance of butterflies in urban fringe areas may be higher due to the more natural environment of suburban areas; (3) Urbanization may lead to differences in the composition of butterfly communities and functional groups because different butterflies have different adaptations to different levels of urbanization.

## 2. Study Areas and Methods

### 2.1. Study Areas

Fuzhou, situated in the southeast coastal region of China, is a highly urbanized city in the subtropical monsoon climate zone. It is positioned between 25°15′ and 26°39′ north latitude and 118°08′ and 120°31′ east longitude. Fujian Province is renowned for its biodiversity and is home to the third-highest number of butterfly species in China, surpassed only by Yunnan and Hainan Provinces [19]. Serving as the capital city of Fujian Province, Fuzhou exemplifies significant urbanization, with a remarkable 87.13% of the built-up area concentrated in the city center (within the first ring road), making it an ideal location for studying the impacts of urbanization on butterfly diversity. The selection of urban waterfront green spaces in Fuzhou was primarily based on two main criteria: (1) The selection of waterfront parks situated in different locations along the city’s river corridors; and (2) The selection of waterfront parks with a variety of vegetation structural compositions and habitat types. Using the aforementioned criteria, we selected ten urban waterfront green spaces along the Minjiang River in Fuzhou. These include Jiangxin Park (4.83 ha), Cangxia Park (3.1 ha), Minjiang Park North Park (44.24 ha), Wenshanli Park (2.30 ha), Guoguang Park (3.1 ha), Minjiang Park South Park (27.4 ha), Huahai Park (65 ha), Dongjiangbin Park (49.19 ha), Nanjiangbin Park (80 ha), and Minjiang River Estuary Wetland Park (281 ha). All selected urban waterfront green spaces are evenly distributed within the administrative boundaries of Fuzhou City (Figure 1).

### 2.2. Classification of Urban Ecological Gradient Type

The method of using urban gradient division was re-evaluated based on previous studies (classification based on built-up area and population data) [20]. We synthesized four approaches to classify urban ecological gradients. These included considerations of Fuzhou’s urban ring road planning (the 2nd loop, the 3rd loop, and the Fuzhou bypass highway), the completion time of the parks (defined as the time when the parks are open), the distance of the parks from the ancient city center of Fuzhou (measured from the parks’ center point to Three Square and Seven Alleys, the ancient city center of Fuzhou) [21], and the built-up area ratio of the city areas used to classify the urban areas into four urban ecological gradients [22] (Table 1).

### 2.3. Sampling Criteria and Systematic Sampling

Firstly, 100 m transects (1 transect for every 2 ha) were established for green spaces using the Urban Biodiversity APP, which was based on the park sizes. Up to 15 transects were selected for each park and evenly distributed within the parks, with a minimum distance of more than 100 m between transects to reduce spatial autocorrelation. The number of transects set up for each park were as follows: Jiangxin Park (3 transects), Cangxia Park (5 transects), Minjiang Park North Park (15 transects), Wenshanli Park (2 transects), Guoguang Park (2 transects), Minjiang Park South Park (14 transects), Dongjiangbin Park (15 transects), Nanjiangbin Park (15 transects), and Minjiang River Estuary Park (15 transects). Subsequently, we employed the proportional sampling method of systematic sampling, based on the green space area, to randomly select parks’ sampled transects from different urban ecological gradient types. Each gradient had a total of 10 transects. The distribution of transects in each urbanization type was as follows: city center areas (Jiangxin Park *n* = 1, Cangxia Park *n* = 2, Minjiang Park North Park *n* = 7); urban areas (Wenshanli Park *n* = 1, Guoguang Park *n* = 1, Minjiang Park South Park *n* = 8); suburban areas (Huahai Park *n* = 4, Nanjiangbin Park *n* = 3, Dongjiangbin Park *n* = 3); and exurban areas (Minjiang River Estuary Wetland Park *n* = 10) (Figure 2).

### 2.4. Butterfly Survey and Functional Classification

Prior to the official survey, we conducted two rounds of pre-tests in April and May 2022 at ten urban green spaces to ensure accurate butterfly species identification during the official survey. The official survey took place from 1 July to 30 September 2022, which corresponds to the most active period for butterfly activities in Fuzhou. Data collection from each park occurred on sunny days with high visibility, low wind speed, and no precipitation. To maximize data accuracy, the survey was conducted from 9:00 a.m. to 12:00 p.m. and from 3:00 p.m. to 6:00 p.m., avoiding the midday period when butterflies tend to hide. Butterfly data collection primarily occurred along 100 m long, 5 m wide transects, and each transect was surveyed for a standard 10 min until no new species were observed. Three survey rounds were conducted (July to September), with one round each month. The survey was mainly conducted by three observers with professional training in ecology, possessing a broad understanding of butterfly habits, behavioral characteristics, and host plants. The capture method was employed for uncertain butterfly species by retaining them in sulfuric acid triangular bags and returning them for manual comparison and identification. The primary reference books were ‘Classification and Identification of Chinese Butterflies’ [23]. 

We classified the butterflies into two groups based on their dietary habits and vertical spatial activity. The feeding habits were categorized into five groups: plant-feeding monophagous butterflies, plant-feeding oligophagous butterflies, plant-feeding polyphagous butterflies, carnivorous butterflies, and scavenging butterflies [24]. In terms of vertical spatial activity, the butterfly populations were divided into three categories: lower spatial activity (h < 0.5 m), middle spatial activity (0.5 m < h < 2.5 m), and upper spatial activity (h > 2.5 m), referring to relevant classification criteria and practical observations [25,26] (Appendix A).

### 2.5. Data Analysis

Firstly, we conducted an analysis of α-diversity and β-diversity among butterfly communities across various urban ecological gradients. Secondly, we explored differences in functional diversity and finally identified indicative species for different ecological gradients [27]. Prior to conducting further analyses, we evaluated the normal distribution of butterfly species richness and abundance data for each urbanization type using the ‘mvShapiroTest’ package in R 4.2.1 [28]. We measured species richness as the number of butterfly species observed at each urbanization and species abundance as the number of individuals observed at each urbanization. Our analysis found that the butterfly’s richness (*p*-value = 0.5344, W = 0.91954) and abundance (*p*-value = 0.3036, W = 0.87149) showed that the normal distribution hypothesis cannot be rejected (*p* > 0.05). Therefore, there was no need to apply logarithmic and square transformations to the data. All statistical analyses were performed using R 4.2.1.

#### 2.5.1. α-Diversity Analysis

α-diversity is a measure of species richness in a given habitat or ecosystem [29]. It is a measure of biodiversity at a local scale, and it can be used to compare the diversity of different habitats or regions. Our study aimed to assess butterfly diversity across various urbanizations, including species richness, abundance, and the Chao1 index for groups with different diets and vertical activity spaces. We used the Shannon index to describe the α-diversity of butterfly communities [30] and calculated it using the ‘vegan’ package [31]. In addition, we analyzed the Chao1 index, which estimates the predicted number of species, with higher Chao1 index values indicating higher species diversity in the community [32]. We also conducted the Shapiro–Wilk test. The *p*-values for the Shannon index (*p*-value = 0.2809, W = 0.86564) and the Chao1 (*p*-value = 0.5182, W = 0.91663) were greater than 0.05, suggesting that the normal distribution hypothesis cannot be rejected. We also employed the ‘iNEXT’ package to calculate species rarefaction and extrapolation curves, enabling us to assess and predict species richness and abundance changes within the community [33]. When the rarefaction curve becomes gradually flattened with increased species abundance, it indicates that the observed species diversity will stabilize gradually, and as more individuals are sampled, fewer new species are expected to be added to the dataset [34] (Appendix A). This approach is crucial in biodiversity surveys as it helps determine if the sample size collected is sufficient to represent the entire butterfly community.

In this study, First, we employed a general linear regression model (GLM) with a Poisson error structure to investigate the impact of different urban ecological types on butterfly diversity [35]. The independent variables included different urbanization types, while the butterfly Shannon index served as the dependent variable. To examine the impact of varying urbanization types on butterfly community responses, we employed a quasi-Poisson distribution regression by fitting general regression linear models (GLM) using the ‘lmer’ function from the ‘lme4’ package. Secondly, we check for multicollinearity between different groups of urban gradients by calculating the variance inflation factor (VIF) [36]. Thirdly, we demonstrate the applicability of our model primarily by residuals to test for model fitness, overdispersion, and homoscedasticity. Finally, we examined the variation in the impact of different urbanization types on butterfly species using analysis of variance (ANOVA). Subsequently, we conducted Tukey post hoc tests with the general linear hypothesis test procedure available in the ‘multcomp’ package [37]. All statistical analyses were performed using R 4.2.1.

#### 2.5.2. β-Diversity Analysis

β-diversity is a measure of dissimilarity in species composition between different habitat communities along an environmental gradient or the rate of species turnover along an ecological gradient [38]. In this study, β-diversity was used to quantify the compositional variation of species under different urban gradients. We calculated β-diversity using a Bray–Curtis similarity matrix based on a log (X + 1) transformation of the abundance data, as shown in the following formula:$$BC_{ij}=1-\frac{2C_{ij}}{s_{i}+s_{j}}$$

Here, *C_ij_* represents the sum of the minimum values of the common butterfly species between two groups, *i* and *j*. *S_i_* and *S_j_* denote the total numbers of butterfly species in groups *i* and *j*, respectively.

To assess the effect of urbanization on species differences, we used non-metric multidimensional scaling (NMDS) to analyze the composition of butterfly communities in urban waterfront green spaces [39]. We also calculated the stress value using the ‘metaMDS’ function, with a stress value of generally <0.2, indicating a significant difference in the models [40]. In addition, we conducted grouped difference comparisons, primarily using the Bray–Curtis distance in Permutational Multivariate Analysis of Variance (PERMANOVA) [41]. For PERMANOVA, we performed multivariate variance analysis with 999 permutations to assess differences in butterfly communities among different urban gradients [42]. All computations were conducted using R 4.2.1.

#### 2.5.3. Functional Diversity and Indicator Species of Butterflies

Functional diversity was analyzed to investigate the overlap and redundancy of butterfly function, and we examined the association between Shannon index and abundance in dietary groups and vertical activity space groups across the four gradients [43]. We used scatter plots from the ‘ggplot2’ package to visualize these relationships [44]. We used species indicator values to assess how butterfly habitats have changed over the course of urbanization. Indicator species effectively explain several variables with ecological characteristics that reflect environmental conditions and the quality of different urbanization types [45]. To combine average species abundance with the probability of species occurring in different urbanization gradients, we used the ‘indicspecies’ package [46]. The package calculates the indicative value (IV) of each species and evaluates the statistical significance of the correlation using a *p*-value threshold of <0.05 *. Using this approach, our study identified the species most strongly associated with each level of urbanization, providing insights into the effects of urbanization on butterfly diversity. A high index value for a species indicates that the species exhibits a higher average abundance in a particular sample group compared to other sample groups (specificity) and is prevalent in most sample groups within that group (evenness). All statistical analyses were performed using R 4.2.1.

## 3. Results

### 3.1. Overview

Between July and September 2022, we conducted butterfly monitoring surveys in 10 urban waterfront green spaces in Fuzhou City. In the sampled transects (which consisted of a mean of 40 systematically sampled transects), a total of 1163 butterflies belonging to 28 species, 6 families, and 19 genera were recorded. Notably, among these, *Kallima inachus* is classified as a second-grade national protected animal in China. The results indicate that the three most commonly observed species were *Pseudozizeeria maha* (327 individuals), *Catopsilia Pomona* (295 individuals), and *Pieris rapae* (167 individuals) (Figure 3). All three species belong to the plant-feeding polyphagous group. But the most common species in the four different urban ecological gradients are as follows: city center areas—*Pseudozizeeria maha* (172 individuals), urban areas—*Catopsilia Pomona* (148 individuals), and suburban areas—*Catopsilia pomona* (81 individuals) and *Argynnis hyperbius* (66 individuals) (Figure 4). Additionally, we observed an increasing trend in both the Shannon diversity index and butterfly richness from city center areas to exurban areas. However, it is worth noting that the number of butterflies from city center areas (353 individuals)–exurban areas (168 individuals) tends to increase and decrease. Finally, we found that the abundance of plant-feeding polyphagous butterflies gradually reduced from the city center (342 individuals)–exurban areas (93 individuals). In comparison, the abundance of plant-feeding oligophagous butterflies gradually increased (11 individuals vs. 70 individuals). The proportion of the two groups of butterflies gradually equalized, and we only found plant-feeding monophagous butterflies in areas on the city’s edge. However, we did not find a clear trend with regard to the vertical space type (Appendix A).

### 3.2. Alpha-Diversity Varies with Urbanization Ecological Gradients

The GLM model showed that urbanization had a significant effect on all butterfly’s Shannon index (*p* = 0.041 *), richness (*p* = 0.001 *), and abundance (*p* = 0.001 *) in terms of urbanization’s impact on butterflies all over. In terms of the effect of urbanization on diet groups, urbanization has a significant impact on the Shannon index (*p* = 0.008 *), richness (*p* = 0.043 *), and abundance (*p* = 0.001 *) of plant-feeding oligophagous butterfly species and on the richness (*p* = 0.001 *) and abundance (*p* = 0.001 *) of the plant-feeding polyphagous group. In terms of the effect of urbanization on butterfly activity space groups, urbanization has a significant impact on the Shannon index (*p* = 0.005 *), abundance (*p* = 0.017 *), and richness (*p* = 0.001 *) of butterflies in the upper activity space group; a significant effect on the richness (*p* = 0.001 *) and abundance (*p* = 0.001 *) of butterflies in the middle activity space group; and a substantial impact on the Shannon index (*p* = 0.001 *) and abundance (*p* = 0.001 *) of butterflies in the lower activity space group. We do not discuss plant-feeding monophagous, carnivorous, and scavenging groups due to the low number of observations (Table 2).

According to the results of the analysis of variance (ANOVA) and Tukey’s post hoc test, the butterfly’s Shannon index (*p* = 0.048 *) and richness (*p* = 0.046 *) differed significantly across urban ecological gradients, and there were pairwise differences between city center–urban and city center–exurban (Figure 5A,B). The Shannon index (*p* = 0.012 *) and richness (*p* = 0.001 *) of the plant-feeding polyphagous group were significantly different across city types, and there was a pairwise difference between city center–urban, urban–exurban, suburban–exurban, and city center–exurban (Figure 6A,B). The abundance of the upper activity space group (*p* = 0.022 *) and the Shannon index of the middle activity space group (*p* = 0.037 *) differed significantly across the different urban types, with pairwise differences in the abundance of the upper activity space group between urban and exurban (Figure 7A), and a significant difference in the Shannon index of the middle activity space group between city center–urban and city center–suburban (Figure 7B). The Shannon index (*p* = 0.021 *) and richness (*p* = 0.045 *) of the lower activity space group were significantly different between the different city types, and there were pairwise differences in the Shannon index in the city center–exurban areas (Figure 8A) and pairwise differences in richness between city center–exurban and suburban–exurban areas (Figure 8B).

### 3.3. Beta-Diversity Varies with Urbanization Ecological Gradients

Our results showed that butterfly communities in urban waterfront green spaces differed significantly between groups at different types of urbanization (stress = 0.1862 < 0.2; R = 0.261; P_adj_ = 0.001 *) (Figure 9A and Figure 10A). This is particularly evident in city center–exurban (R^2^ = 0.224; P_adj_ = 0.002 *), urban–exurban (R^2^ = 0.251; P_adj_ = 0.002 *), and suburban–exurban areas (R^2^ = 0.182; P_adj_ = 0.002 *). Additionally, differences were found in the group of plant-feeding polyphagous butterflies (stress = 0.1612 < 0.2, *p* = 0.001 *) (Figure 9B and Figure 10B). This is particularly evident in city center–exurban (R^2^ = 0.252; P_adj_ = 0.002 *), urban–exurban (R^2^ = 0.305; P_adj_ = 0.002 *), and suburban–exurban areas (R^2^ = 0.237; P_adj_ = 0.002 *). No significant differences were found in other dietary or vertical activity space groups.

### 3.4. Functional Diversity and Butterflies Indicator of Different Urbanization Ecological Gradients

For the diet group, our LRM results indicated that the plant-feeding oligophagous group’s Shannon index significantly affected abundance in urban areas (*p* = 0.026 *, Figure 10A) and the exurban regions (*p* = 0.022 *, Figure 10B); in city center areas, the plant-feeding polyphagous group of α-diversity significantly influenced abundance (*p* = 0.006 *, Figure 10C). In terms of vertical activity space groups, LRM results showed that in exurban areas, the Shannon index of the upper activity space group significantly influenced abundance (*p* = 0.027 *, Figure 10D); in urban areas and suburban areas, the Shannon index of the middle activity space group significantly influenced abundance (*p* = 0.021 *, Figure 10E/*p* = 0.002 *, Figure 10F); and in urban areas, the Shannon index of the lower activities space group greatly affected abundance (*p* = 0.001 *, Figure 10G) (Appendix A). 

Furthermore, we identified five indicative species, with one indicative species screened in the city center areas (*Pieris rapae*: IV = 0.377, *p* = 0.019 *), urban areas (*Catopsilia pomona*: IV = 0.502, *p* = 0.007 *), and suburban areas (*Eurema hecabe*: IV = 0.563, *p* = 0.004 *), and two species in the exurban areas (*Chilades pandava*: IV = 0.714, *p* = 0.001 *, Papilio xuthus: IV = 0.300, *p* = 0.049 *) (Figure 11) (Appendix A).

## 4. Discussion

### 4.1. Urbanization Affects Butterflies’ Diversity in Urban Waterfront Green Spaces

Urbanization greatly affects the Shannon index, richness, and abundance of butterfly communities along an urban ecological gradient. Consistent with the findings of Sing et al. (2019), we equally demonstrated that urbanization significantly and negatively impacts butterfly diversity, reducing the structure of butterfly communities, according to NMDS diagrams (Figure 9A) [47]. However, it is worth noting that we found a positive effect of urbanization on butterfly abundance, which differs from the majority of studies demonstrating a negative impact of urbanization on butterfly abundance [5,48,49]. Urbanization likely increases the abundance of flowering plants, providing a richer nectar source to support individual butterflies [50]. However, most of these urban flowering plants are exotic ornamental species that not all butterflies can feed on, which forces some butterfly species to stay away from the city, especially some of the large specialist butterflies [26]. Small- or medium-sized generalists may occupy these empty ecological niches, leading to a decrease in butterfly diversity and an increase in abundance. Moreover, we found that suburban areas had the highest Shannon index and urban areas had the highest abundance, which is consistent with the conclusion of the moderate disturbance hypothesis that more complex spatial patterns (heterogeneity) can provide a habitat for more species [51]. Additionally, previous studies have shown that highly urbanized areas may favor species with faster growth rates and increased resistance to exotic species, resulting in the dominance of specific species and populations, leading to significant differences in species and population composition between urbanization areas [52,53]. We found that butterflies of the families Pieridae and Lycaenidae became the dominant urban populations, and most of these butterfly populations belonged to plant-feeding polyphagous butterflies. Finally, we found significant differences between butterfly communities and plant-feeding polyphagous butterfly communities in the four urbanized areas, mainly in the exurban area. This may be a result of differences in butterfly populations due to differences in landscape features caused by urbanization [54], while no differences were found in the other communities.

### 4.2. Urbanization Affects Butterflies’ Diet Groups and Activity Spaces Groups in Urban Waterfront Green Spaces

We observed the highest number of individuals of plant-feeding polyphagous butterflies and the lowest number of specific butterflies exclusively found in natural areas farther away from city centers, such as suburban and exurban areas. These findings are consistent with Bergerot’s (2010) research, which revealed that specialist butterflies tended to avoid urban areas more strongly than diet generalist butterflies [53]. Furthermore, we observed a gradual decrease in the abundance of plant-feeding polyphagous butterflies along the urbanization gradient from urban to natural areas. In contrast, the abundance of plant-feeding oligophagous butterflies showed a gradual increase. These findings are in line with the research by Palash (2022), suggesting that butterflies with narrow diets are more vulnerable to the effects of urbanization compared to those with broader diets. This again highlights the significant impact of urbanization on the survival of butterflies from different dietary functional groups, with polyphagous butterflies being more likely to thrive in urban environments [26]. In addition, we found that urbanization significantly affected the abundance of butterflies in different activity spaces. Differences in butterfly function may stem from differences in plant type occupancy due to urbanization. Zhang (2012) found that the proportion of herb layers gradually decreased with urbanization, while the proportion of shrub and tree layers first increased and then decreased [55]. Butterflies and plants are a co-evolving, mutually supportive relationship. These changes in landscape character due to urbanization may affect the richness and abundance of butterflies in different activity spaces.

### 4.3. Urbanization Affects Butterfly Functional Diversity and Butterfly Indicators in Urban Waterfront Green Spaces

We found significant differences in the functional diversity of butterflies in urban waterfront green spaces across four urban ecological types. In city center areas, there was a positive correlation between the abundance of the plant-feeding polyphagous butterfly and the Shannon index, suggesting that city center areas may provide richer food sources and more suitable habitat conditions for the survival of butterflies in this group. This aligns with Tew et al. (2021), who reported that urban areas have a more diverse nectar supply, mainly provided by non-native flowering plants, making generalist butterflies more likely to thrive in urban environments [56]. We also found a positive correlation between butterfly abundance in the plant-feeding oligophagous group and the Shannon index in urban and exurban areas. This could be attributed to the more complex plant resources and habitats in these areas that attract butterflies of the plant-feeding oligophagous group. These findings are consistent with Lin et al.’s study, suggesting that moderately urban-disturbed areas and areas with low levels of urbanization may support a more diverse specialist butterfly community [48]. Regarding the vertical activity space, we observed a significant positive correlation between the Shannon index and abundance in the middle and lower space activities group in urban areas and a similar positive correlation in the middle space activities group in suburban areas. This indicates that urban areas have sufficient resources to support the middle and lower space activities group, and there may not be a competitive relationship between them, leading to specific functional overlaps. This observation aligns with Zhang et al. (2012), who reported a gradual decrease in herbaceous plants and a gradual increase in trees and shrubs from urban to natural areas [55]. The high density of the herbaceous layer in urban areas provides a suitable habitat for butterflies active in the middle and lower activity spaces, such as the Pieridae and Lycaenidae families. Additionally, we found a significant positive correlation between the Shannon index and abundance in the upper space activities group in exurban areas. This might be due to the more natural habitats in exurban areas, which favor larger butterflies with more potent flight abilities that require higher ecological quality. These findings are consistent with Reeder et al. (2005), who reported a positive correlation between vegetation height and vertical density and the abundance of large butterflies generally active in the upper space [57].

In addition, our study has identified five butterfly species that can potentially be used as indicative species of urbanization trends in urban waterfront green spaces in Fuzhou. Pieris rapae (city center areas), Catopsilia Pomona (urban areas), Eurema hecabe (suburban areas), Chilades pandava, and Papilio Xuthus (exurban regions). The Indicative Value (IV) index identifies indicative species and assesses their association with particular environmental conditions. Among the above five butterflies, Pieris rapae, Catopsilia pomona, and Eurema hecabeare all in the Pieridae family and mainly move in the lower and middle space as plant polyphagous butterflies. This implies that plant-feeding polyphagous butterfly groups are better equipped to adapt to different urbanization impacts, as well as exhibit greater adaptability in their responses to urbanization.

### 4.4. Limitations and Perspectives

The main limitations of this study are as follows: (1) Data were collected solely from urban waterfront green spaces, and the limited types of urbanization studied may only represent part of the urban butterfly community; (2) This study focused only on the overall impact of urbanization on butterfly diversity, without fully considering factors such as climate change, land cover type [58], ecological corridor connectivity [59], nectar plants [60], host plants [61,62], predators, and competitors [63]. (3) This study primarily concentrated on the summer season, potentially missing out on capturing seasonal and annual changes in butterfly populations. To further enhance our understanding, we recommend the following future developments: (1) Expand the sampling types and locations to include mountain parks [64], comprehensive parks, waterfront parks, forest parks [65], and conservation areas [66], covering a more comprehensive range of habitat classes; (2) Provide a more detailed delineation of the urban gradient to increase the study’s precision and complexity; (3) Implement long-term butterfly population monitoring to gain a comprehensive understanding of seasonality and population trends; (4) Quantify climatic data, nectar and host plant data, land cover type, and connectivity data using GIS [67,68], RS [68], and semantic transaction of images [69] to conduct in-depth investigations of the positive and negative impacts of urbanization on butterfly diversity; (5) Study the interactions between butterflies and their predators or competitors to deepen our understanding of butterfly community dynamics in urban environments.

## 5. Conclusions

In this study, we divided 10 urban waterfront green spaces into 4 different urban ecological gradient types and examined the differences in butterfly communities and functional group differences under different urban ecological types. Based on the analyses, we summarized the following four main conclusions: (1) Urbanization has led to a decrease in the Shannon index and richness of butterflies, and while there are more individual butterflies in city centers and urban areas, it has generally led to a simpler butterfly community structure; (2) Urbanization has significant adverse effects on specialist butterflies than generalist butterflies, with plant-feeding polyphagous butterflies better adapted to urbanization than plant-feeding oligophagous butterflies; (3) Urban areas and exurban areas supported more functional groups of butterflies, and there was a more functional overlap; (4) *Pieris rapae*, *Catopsilia pomona*, *Eurema hecabe*, *Chilades pandava*, and *Papilio xuthus* can be used as indicator species to monitor the diversity of butterflies in urban waterfront parks, of which the first three urban indicator species belong to Pieridae. 

To promote urban species diversity, in addition to creating new green spaces, expanding the area of green spaces, and reducing urbanization disturbances, we recommend improving the ecological quality of urban waterfront green spaces and increasing the diversity of native plant species and the heterogeneity of habitat types within green spaces, focusing on the vertical spatial proportion of trees, shrubs, and herbs to improve biodiversity in highly urbanized areas.

## Figures and Tables

**Figure 1 insects-14-00851-f001:**
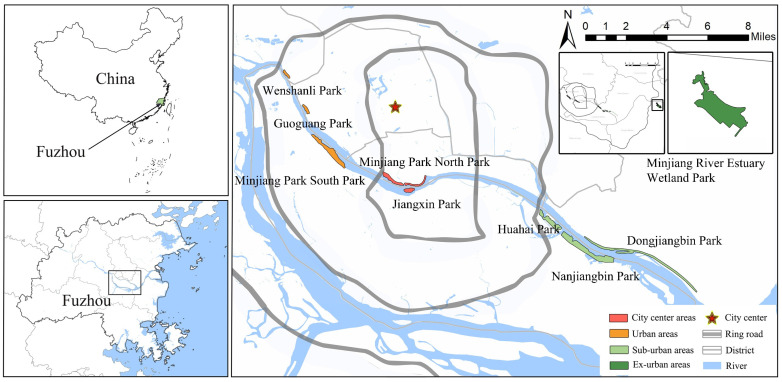
This figure shows the geographical location of the distribution of 10 urban waterfront green spaces at 4 different urban ecological gradients within Fuzhou City.

**Figure 2 insects-14-00851-f002:**
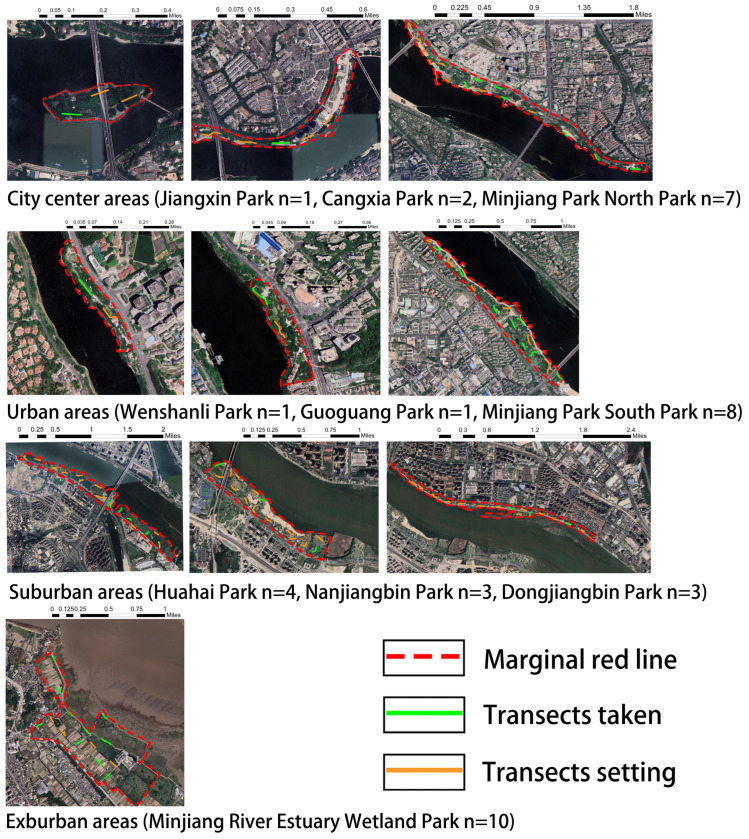
Transects taken and location of 10 parks.

**Figure 3 insects-14-00851-f003:**
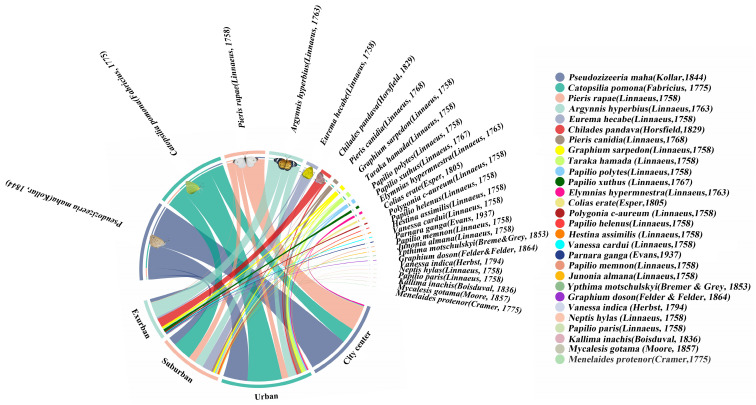
Chord diagrams of different urban ecological gradients and butterfly species, with species names in the upper half-circle and different ecological gradient names in the lower half-circle. The connecting lines indicate butterfly species present within the urban gradient, with thicker line segments showing more individuals of the species.

**Figure 4 insects-14-00851-f004:**
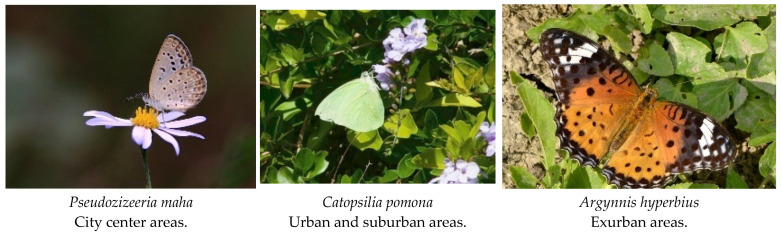
Butterfly species with highest abundance in four urbanization types.

**Figure 5 insects-14-00851-f005:**
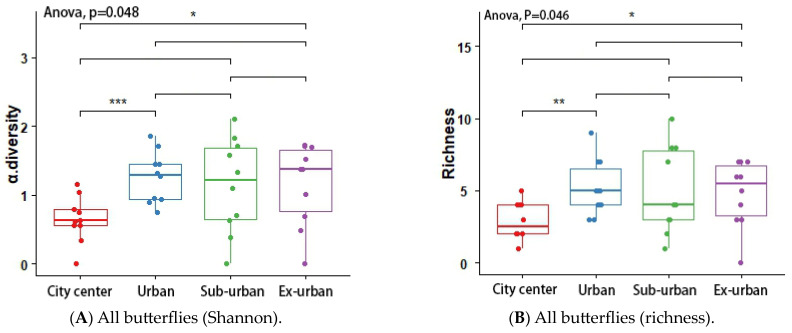
Butterflies’ Shannon index (**A**) and richness (**B**) in urban waterfront green spaces across various levels of urbanization. Box plots show the data spread for median values and use hollow circles to represent outliers and solid circles to represent means. Significance levels for paired differences are indicated by asterisks (*** *p* < 0.001; ** *p* < 0.01; * *p* < 0.05).

**Figure 6 insects-14-00851-f006:**
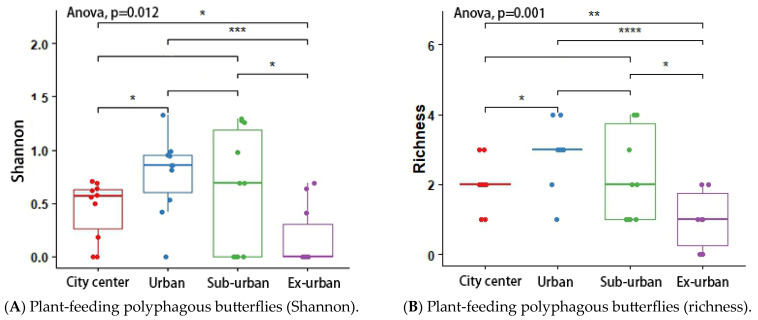
Plant-feeding polyphagous butterflies’ Shannon index (**A**) and richness (**B**) in urban waterfront green spaces across various levels of urbanization. Box plots show the data spread for median values and use hollow circles to represent outliers and solid circles to represent means. Significance levels for paired differences are indicated by asterisks (**** *p* < 0.0001; *** *p* < 0.001; ** *p* < 0.01; * *p* < 0.05).

**Figure 7 insects-14-00851-f007:**
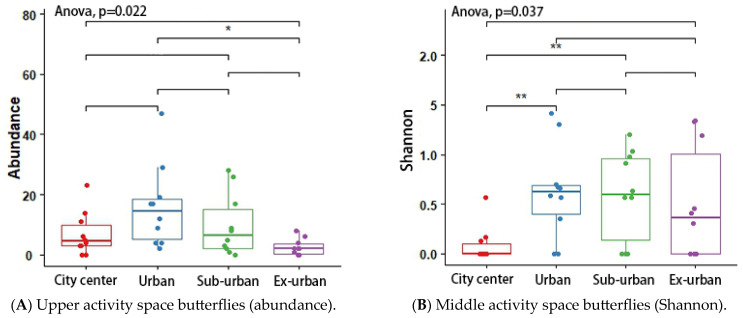
Upper activity space butterflies’ abundance (**A**) and middle activity space butterflies’ Shannon index (**B**) in urban waterfront green spaces across various levels of urbanization. Box plots show the data spread for median values and use hollow circles to represent outliers and solid circles to represent means. Significance levels for paired differences are indicated by asterisks (** *p* < 0.01; * *p* < 0.05).

**Figure 8 insects-14-00851-f008:**
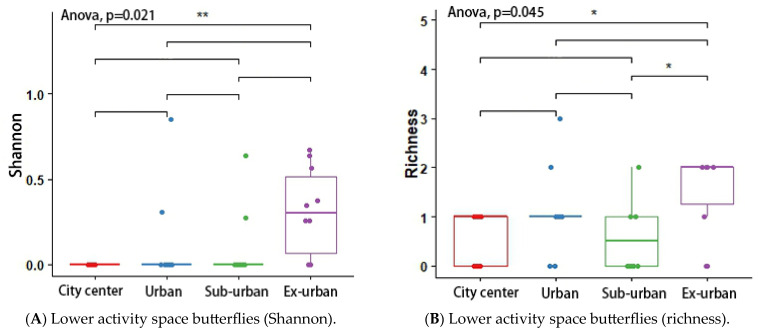
Lower activity space butterflies’ Shannon index (**A**) and richness (**B**) in urban waterfront green spaces across various levels of urbanization. Box plots show the data spread for median values and use hollow circles to represent outliers and solid circles to represent means. Significance levels for paired differences are indicated by asterisks (** *p* < 0.01; * *p* < 0.05).

**Figure 9 insects-14-00851-f009:**
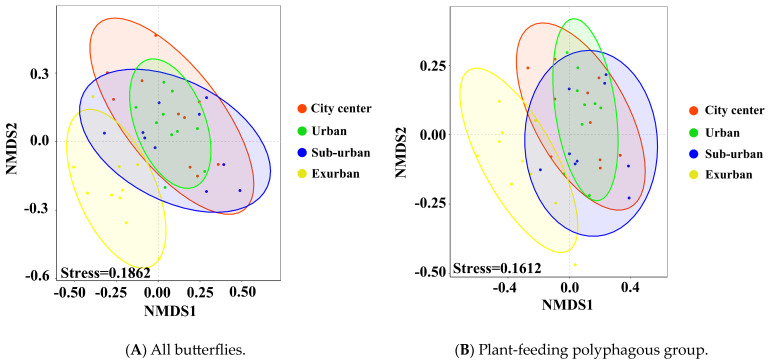
NMDS two-dimensional maps of the all butterfly community (**A**) and the plant-feed ing polyphagous butterfly group (**B**) from 40 sample transects (10 urban center areas = red circles, 10 urban areas = green circles, 10 suburban areas = blue circles, and 10 exurban areas = yellow circles). Stress values were 0.1862 and 0.1612 (stress <0.2) for the overall butterfly community and the plant-feeding polyphagous butterfly group, respectively.

**Figure 10 insects-14-00851-f010:**
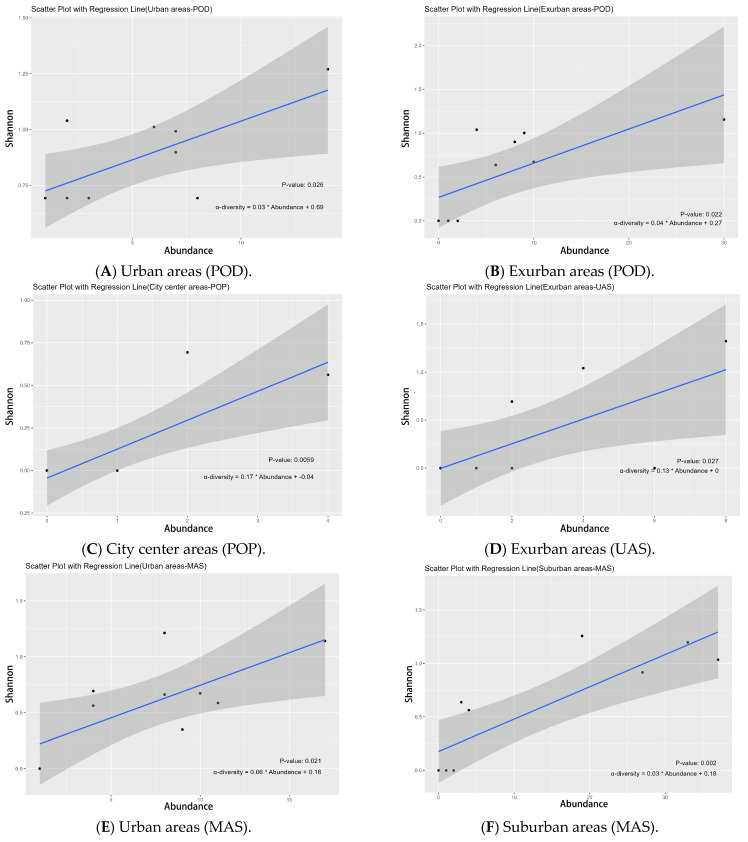
Linear regression model of Shannon index and abundance for dietary group and vertical activity space groups across urbanization types. The abbreviations used in the figure have the following meanings: POD indicates the Shannon index of plant-feeding oligophagous butterflies, POP indicates Shannon index of plant-feeding polyphagous butterflies, UAS indicates Shannon index of upper activity space, MAS indicates Shannon index of middle activity space, LAS indicates Shannon index of lower activity space, and SA indicates species abundance.

**Figure 11 insects-14-00851-f011:**
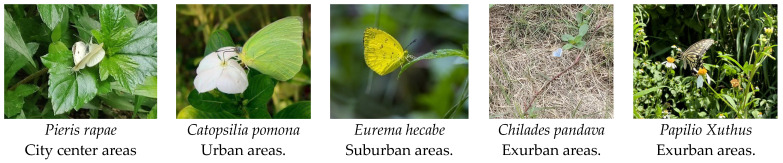
Five indicator species for different urban ecological gradients.

**Table 1 insects-14-00851-t001:** Classification of study areas into 4 urban ecological gradients based on ring road planning, distance to the city center, time to park completion, and built-up area ratio.

Ecological Gradient Types	Park Introduction	Ring Road (Loop)	Completion Time (Mean)	Building Density	Distance from the City Center (Mean)
City center area	JXP (1982) (3.42 km) CXP (1985) (3.25 km) MJPNP(2001) (3.26 km)	1st–2nd loop	1989	76.81%	3.31 km
Urban area	MJPSP (2002) (3.52 km) GGP (2004) (4.55 km) WSLP (2020) (5.79 km)	2nd–3rd loop	2008	62.34%	4.62 km
Suburban area	HHP (2013) (7.16 km) DJBP (2013) (13.74 km) NJBP (2020) (11.09 km)	3rd—Fuzhou Bypass Highway	2015	48.71%	10.66 km
Exurban area	MJREWP (2021) (32.86 km)	Outside Fuzhou Bypass Highway	2021	——	32.86 km

**Table 2 insects-14-00851-t002:** Test results of the effect of urbanization on the butterfly diet habit and vertical activity space under the general linear model (GLM) fitted Poisson model. * *p* < 0.05 means that urbanization has a significant effect on the parameter.

Data Type	Diet and Space Level	Parameter	Estimate	Std. Error	*p*-Value	Df
All butterflies	——	Shannon index	0.230	0.141	0.041 *	36
		Richness	1.649	0.142	0.001 *	
		Abundance	3.622	0.341	0.001 *	
Diet habit group	Plant-feeding monophagous butterfly	Shannon	−22.300	5561.000	0.997	36
		Richness	−21.300	4479.000	0.996	
		Abundance	−21.300	4869.000	0.997	
	Plant-feeding oligophagous butterfly	Shannon index	−1.052	0.374	0.008 *	36
		Richness	0.531	0.253	0.043 *	
		Abundance	1.629	0.379	0.001 *	
	Plant-feeding polyphagous butterfly	Shannon index	−0.262	0.210	0.222	36
		Richness	1.065	0.129	0.001 *	
		Abundance	3.398	0.361	0.001 *	
	Carnivorous butterfly	Shannon index	−26.300	0.333	0.001 *	36
		Richness	−1.609	0.486	0.002 *	
		Abundance	0.470	0.483	0.337	
	Scavenging butterfly	Shannon index	−26.300	0.333	0.001 *	36
		Richness	−21.300	5730.000	0.997	
		Abundance	−20.300	4256.00	0.996	
Vertical space activities group	Upper space activities	Shannon index	−1.385	0.465	0.005 *	36
		Richness	0.531	0.211	0.017 *	
		Abundance	2.773	0.227	0.001 *	
	Middle space activities	Shannon index	−0.471	0.262	0.081	36
		Richness	0.875	0.198	0.001 *	
		Abundance	2.163	0.435	0.001 *	
	Lower space activities	Shannon index	−2.157	0.542	0.001 *	36
		Richness	0.095	0.243	0.697	
		Abundance	2.542	0.583	0.001 *	

## Data Availability

The data used to support the findings of this study are available from the corresponding author upon request.

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
