# Peer review of "The Impact of Urbanization on Taxonomic Diversity and Functional Similarity among Butterfly Communities in Waterfront Green Spaces"

_insects, 2023, doi:10.3390/insects14110851_

Round 1

Reviewer 1 Report

Comments and Suggestions for Authors

Author presented an investigation on the impact of urbanization on butterfly taxonomic and functional diversity. They surveyed 10 urban waterfront green spaces in Summer 2022 investigating three main aspects: 1) effect of different urban ecological gradients on butterfly communities; 2) differences between butterfly communities among urban ecological gradients; 3) differences in the response of butterfly functional groups to different urban ecological gradient areas.

I found this work of interest. It can give many insights for future study and provide suggestions for city management plans.  

In my opinion, the introduction provide sufficient background and the research design is appropriate. Results are cleary presented and and properly discussed. 

I have just very minor comments:

- Figures 1, 2, and 4 are very difficult to read due to the size of the characters. The lettering is very poorly visible. If possible, enlarge the font size. In Figures 9, 10 and 11 the captions are also unreadable for the same reason.

- Figure 2: the legend can only be placed once, avoiding repetition in all map images. The lines delimiting the segments in the maps are really small and thin and can hardly be seen. 

- Line 48: "...it also alters urban population densities..." What does 'it' refer to? Please, clarify.

- It is not immediately clear that the 'segments' are not the transects (lines 129-146). I would recommend explaining straight away that within the segments are transects where the butterflies were actually sampled. 

Author Response

Dear Professor.

We are very grateful to the reviewers for their comments, and we have made revisions based on the suggestions. Responses, manuscripts, and attachments are now uploaded. 

Wish you good health and happy life!

Reviewer 2 Report

Comments and Suggestions for Authors

This paper examines an interesting issue – the role that parks play in urban conservation.  The authors did a great job of characterizing the butterfly communities at their sample sites.  However, the paper falls short in several areas.  On a basic level, I suggest that the authors find the basic essence of their research, cut the paper down to that essence, and cut out most of the speculative stuff.  Overall, my take is that the authors try to extrapolate too broadly from a basic kernel of data.  The paper should be reduced in scope, rewritten, and re-submitted.

Here are some general observations.  I’ve attached a pdf file with comments for the authors to see as well.

Other than categorize the sample sites relative to their location in an urban zone, the authors provide little insight into the other differences between the sites that would help explain the differences they observed.  For example, no mention of preserve size is included – larger generally equals high alpha – diversity.  Edge to volume ration could explain how deeply “urban pressure” reaches into each site.  Distance between the sites could be used to explore the beta-diversity trends observed.  Likewise, for distance from the nearest block of “non-urban” habitat.  All these measures can still be obtained from GIS analysis. And would be highly informative about the nature of urban stressors on butterfly communities.

The differences in habitat quality are harder to collect post study.  For example, instead of referring to a paper that says urban habitat support more flower species than non-urban habitats, it would have been nice to simply have counted the number of flower species observed at each site.  The reality is, you have no idea if this is true for your study.  Just a paper that says that in a different region, that’s a trend they a observed.  But you discuss this as if it were an established truth for your study.   Bottom line -  something could have been assessed to provide better insight into how the urban gradient plays out from a habitat standpoint. 

I’m not too sure the authors understand alpha- and beta-diversity.  These are both very broad terms that include multiple measures.  Species richness is a measure of alpha-diversity.  Likewise, the Shannon index is a measure of alpha-diversity.  There are many other measures as well.  So – refer to measures of Shannon indices by name.  Similarly, beta-D is between sites – and there are tests that can actually tell you which sites are significantly different  - I will wager that exurban habitats  are different from the group “city center+Urban+sub-urban”.  Use a SIMPROF test to do this.

The figures are illegible.  At 200%, I can’t read figure 4.  Others figures are almost just bad.

The figure legends should be improved so the reader can understand the main points without referring back to the text.

The authors do a great job of recognizing the limitations of this paper in lines 462-466.  My conclusion is that they should re-write the manuscript with these limitations in mind.  This will produce a much shorter paper, but one that is more likely to be used by other researchers.

Comments on the Quality of English Language

There are issues of confusion that could be resolved if a fluent translator between english and the native language  were to help with the editing.  

Author Response

(The authors gave the same response as above.)
